# How to Determine the Preferred Image Distribution of a Black-Box Vision-Language Model?

## Abstract

Large foundation models have revolutionized the field, yet challenges remain in optimizing multi-modal models for specialized visual tasks. We propose a novel, generalizable methodology to identify preferred image distributions for black-box Vision-Language Models (VLMs) by measuring output consistency across varied input prompts. Applying this to different rendering types of 3D objects, we demonstrate its efficacy across various domains requiring precise interpretation of complex structures, with a focus on Computer-Aided Design (CAD) as an exemplar field. We further refine VLM outputs using in-context learning with human feedback, significantly enhancing explanation quality. To address the lack of benchmarks in specialized domains, we introduce CAD-VQA, a new dataset for evaluating VLMs on CAD-related visual question answering tasks. Our evaluation of state-of-the-art VLMs on CAD-VQA establishes baseline performance levels, providing a framework for advancing VLM capabilities in complex visual reasoning tasks across various fields requiring expert-level visual interpretation. We release the dataset and evaluation codes at `https://github.com/...`.

## 1 Introduction

Large foundation models have revolutionized the AI landscape, providing unparalleled capabilities across various domains [3]. Vision-Language Models (VLMs), a subset of these models, integrate visual and textual information, enabling complex tasks such as image captioning, visual question answering, and multi-modal reasoning [4, 28]. Despite their impressive performance, a significant challenge remains: extracting the most useful knowledge from these black-box models.

Prompt engineering has seen extensive research and application in large language models, optimizing inputs to elicit more accurate and relevant responses [25, 18]. However, the multi-modal nature of VLMs introduces additional layers of complexity. These models must interpret and integrate information from both visual and textual inputs, and the optimal prompting strategy can vary significantly based on the image distribution [19].

Understanding image view distribution is crucial across various domains. In mechanical design, different views of parts and assemblies enhance comprehension of complex structures, aiding design and analysis. In architecture and construction, multiple perspectives of building designs help assess structural integrity and plan activities. In robotics and autonomous driving, diverse viewpoints improve navigation and object manipulation. In surveillance and security, integrating views from multiple cameras enhances monitoring accuracy. In medical imaging, different views of scans like MRI and CT provide comprehensive insights for diagnosing diseases, requiring models to integrate information from various angles.

Submitted to 38th Conference on Neural Information Processing Systems (NeurIPS 2024). Do not distribute.

In this work, we address the challenge of determining which image distributions lead to better outputs from a black-box VLM. Specifically, we focus on scenarios where multiple views of objects are available, such as renderings of images taken under different conditions [29]. Given that we often lack information about the data on which the VLM was trained, and do not have access to the model weights, properties, or gradients, traditional methods for assessing model confidence are not applicable [6].

To overcome this, we propose a novel method to measure the confidence of a VLM without requiring access to its internal parameters. Our approach involves analyzing the outputs produced by the model under different image distributions. By systematically evaluating the model's confidence across various distributions, we can infer the image distributions that the VLM "prefers," leading to more reliable and accurate outputs. Our approach is based on the hypothesis that higher consistency in a VLM's outputs, despite variations in input prompts, indicates higher model confidence. This hypothesis is grounded in the principle that a model with a robust internal representation of the input should produce consistent outputs even when the input is paraphrased. This aligns with recent work on self-consistency in language models [33] and relates to the concept of model calibration [13].

We also apply in-context learning with human feedback (ICL-HF) to refine and improve VLM outputs. By incorporating expert knowledge through iterative feedback, we demonstrate enhancements in the quality and accuracy of VLM-generated explanations for complex 3D mechanical parts. This process provides valuable insights into the learning dynamics of VLMs in specialized domains.

Building upon these methods, we present CAD-VQA, a new dataset specifically designed to evaluate VLMs' understanding of 3D mechanical parts in Computer-Aided Design (CAD) contexts. This dataset, comprising carefully curated images, questions, and answers, addresses a gap in the field by providing a benchmark for assessing VLM performance in specialized technical domains.

The main contributions of this work are:

1. A novel method for measuring VLM confidence based on output consistency across different image distributions, without access to internal model parameters.

2. An application of in-context learning with human feedback (ICL-HF) to improve VLM performance in the specialized domain of 3D mechanical part analysis.

3. CAD-VQA: A new dataset for evaluating VLMs on CAD-related visual question answering tasks, addressing the lack of benchmarks in this domain.

4. Evaluation of state-of-the-art VLMs on the CAD-VQA dataset, establishing baseline performance levels for future research.

While we acknowledge that high consistency in model utputs could potentially result from model biases or limitations, rather than true confidence, we believe our approach provides a valuable proxy for assessing the reliability of (black-box) VLM outputs across different image distributions. Moreover, the combination of our consistency measurement technique, application of ICL-HF, and the CAD-VQA dataset offers a comprehensive framework for advancing the capabilities of VLMs in specialized visual reasoning tasks.

## 2   Related Work

**Prompt engineering** for large language models has been extensively explored, as demonstrated by Reynolds and McDonell [25], Liu et al. [18], and Radford et al. [24]. These studies focus on designing effective prompts to elicit desired responses from language models, thereby enhancing their utility in various applications. Recent works such as Gao et al. [11], Lester et al. [16], Wei et al. [30], and Sanh et al. [26] have further expanded on prompt engineering techniques, introducing methods like prompt tuning and instruction-based learning. However, prompt engineering for multi-modal models remains relatively underexplored, particularly in the context of image distributions and their impact on model performance.

**Prompt engineering for vision-language models.** While much work has been done on prompt engineering for language models [18], the extension to multimodal scenarios presents unique challenges. Cho et al. [7] proposed a unified framework for vision-language prompt learning, demonstrating the potential of tailored prompts in improving model performance.

**The complexity of evaluating black-box models** without access to their internal parameters is a well-known challenge. Tsimpoukelli et al. [29] investigate multimodal few-shot learning with frozen language models, addressing the difficulties in adapting pre-trained models to new tasks with limited data. Similarly, Chen et al. [6] evaluate large language models trained on code, proposing methods to assess model confidence and performance without direct access to model internals. Our work builds on these foundations by addressing the specific challenge of determining preferred image distributions for VLMs. By focusing on scenarios with multiple views of objects, such as renderings under different conditions, we propose a novel approach to measure model confidence and optimize input data for better outputs. This contribution aims to bridge the gap in the existing literature on prompt engineering and evaluation for multi-modal models. Hendricks et al. [14] proposed a probing framework to assess the grounding capabilities of VLMs, highlighting the importance of understanding how these models integrate visual and linguistic information. Similarly, Cao et al. [5] investigated the inner workings of VLMs, providing insights into their decision-making processes.

**The concept of using consistency as a measure of model performance** has gained traction in recent years. Xu et al. [33] demonstrated that self-consistency can improve chain-of-thought reasoning in language models, which aligns with our approach of using consistency to assess VLM outputs. In the context of vision-language tasks, Frank et al. [10] explored the use of consistency in visual question answering, showing how it can be leveraged to improve model accuracy.

**The concept of in-context learning with human feedback**, which we employ in our study, draws inspiration from recent advancements in reinforcement learning from human feedback (RLHF) [8, 27, 22]. While we don't use reinforcement learning directly, the principle of incorporating human feedback to improve model outputs is similar. This approach aligns with broader trends in interactive and iterative learning paradigms [2], as well as methods for fine-tuning language models with human preferences [34, 31]. The integration of expert knowledge through feedback mechanisms has also been explored in various domain-specific applications [32].

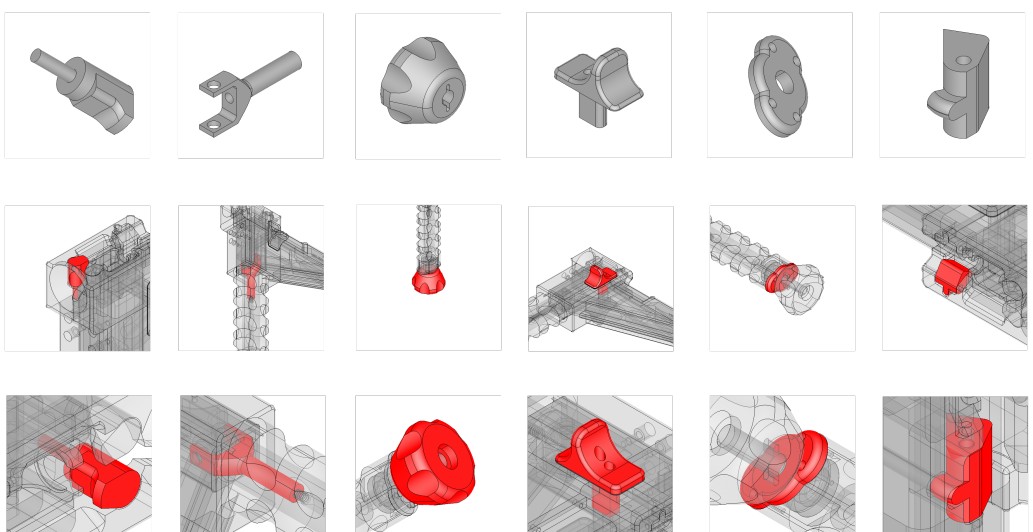

Figure 1: Sample data visualization showing different image distributions and generated explanations. First, second and third row correspond to distributions A, B, and C of the same object, respectively.

## 3   Method

In this work, we address the challenge of determining which image distributions lead to better outputs from a black-box VLM. Specifically, we use GPT-4o [20] for our experiments, but it can simply replaced by any other VLM. GPT-4o is currently known for its state-of-the-art performance in integrating visual and textual information. We focus on scenarios where multiple views of objects are available, such as renderings of images taken under different conditions. Given that we often lack information about the data on which the VLM was trained, and do not have access to the

model weights, properties, or gradients, traditional methods for assessing model confidence are not applicable.

Given $N$ image distributions $\{I_1, I_2, \ldots, I_N\}$, our goal is to determine which distribution leads to better performance when using a black-box Vision-Language Model (VLM), such as GPT-4o [20], where we do not have access to model weights, gradients, or probabilities. To achieve this, we propose a method to measure the consistency of the VLM's outputs across different image distributions.

The underlying hypothesis of this methodology is that higher consistency in the VLM's outputs, despite variations in the textual prompts, indicates higher model confidence. Model confidence refers to the certainty with which a model produces an output given an input. For a VLM, confidence can be understood as the model's ability to generate consistent and reliable outputs despite variations in the input prompts. A robust model should produce similar outputs when presented with semantically equivalent but syntactically different prompts. This robustness indicates that the model has a stable and reliable understanding of the input image, suggesting higher confidence in its outputs.

Let $P$ be the set of paraphrased prompts and $I$ be an image distribution. For a robust and confident model, the outputs $O_{P,I}$ should exhibit minimal variance. Formally, for paraphrased prompts $P_1$, $P_2, \ldots, P_C$, the outputs $O_{I,P_1}, O_{I,P_2}, \ldots, O_{I,P_C}$ should be similar:

$$\text{Var}(O_{I,P_1}, O_{I,P_2}, \ldots, O_{I,P_C}) \approx 0$$

Here, variance (Var) is a measure of inconsistency. Lower variance implies higher consistency, which can be interpreted as higher confidence.

**Prompt paraphrasing.** Given a textual prompt $P$ that aims to extract information from an image, we generate $C = 3$ different paraphrased commands $\{P_1, P_2, P_3\}$ using the chat version of GPT-4 and manually verify them. These paraphrased commands are designed to maintain the same semantic meaning while varying the phrasing. The full prompt used for paraphrasing can be found in Appendix A.1.

After generation, we manually review the paraphrases to ensure they meet our criteria for semantic equivalence and diversity. Any paraphrases that deviate too far from the original meaning or don't provide sufficient variation are replaced with manually crafted alternatives.

**Collecting VLM outputs.** For each image distribution $I_n$ and each paraphrased command $P_c$, we collect the VLM's output $O_{n,c}$:

$$O_{n,c} = \text{VLM}(I_n, P_c)$$

where $n \in \{1, 2, \ldots, N\}$ and $c \in \{1, 2, \ldots, C\}$. This results in a set of outputs $\{O_{n,1}, O_{n,2}, \ldots, O_{n,C}\}$ for each image distribution $I_n$.

### 3.1 Measuring Consistency

We measure the consistency of the outputs for each image distribution using three different methods:

*ROUGE and BLEU Scores.* We calculate the ROUGE [17] score i.e., ROUGE-1, ROUGE-2, ROUGE-L, and BLEU [23] scores for the outputs within each image distribution. Let $S_{n,c}$ be the score between $O_{n,c}$ and a reference output. The consistency score $C_{\text{ROUGE/BLEU},n}$ for image distribution $I_n$ is defined as the average score across all paraphrased commands:

$$C_{\text{ROUGE/BLEU},n} = \frac{1}{C} \sum_{c=1}^{C} S_{n,c}$$

*BERT Embedding Cosine Similarity.* We embed each output $O_{n,c}$ using a BERT model [9] and calculate the cosine similarity between the embeddings. Let $\text{BERT}(O_{n,c})$ be the embedding of $O_{n,c}$. The consistency score $C_{\text{BERT},n}$ for image distribution $I_n$ is defined as the average cosine similarity between all pairs of embeddings:

$$C_{\text{BERT},n} = \frac{2}{C(C-1)} \sum_{i=1}^{C} \sum_{j=i+1}^{C} \cos(\text{BERT}(O_{n,i}), \text{BERT}(O_{n,j}))$$

**GPT-based Consistency Judgement.** We use GPT-4o to act as a judge and provide a consistency score for the outputs within each image distribution. The detailed prompt for consistency judgment is provided in Appendix A.2.

GPT-4o then provides a consistency score between 0 and 1, where 0 means completely inconsistent and 1 means perfectly consistent. Let $G(O_{n,1}, O_{n,2}, \ldots, O_{n,C})$ be the consistency score given by GPT-4o. The consistency score $C_{\text{GPT},n}$ for image distribution $I_n$ is:

$$C_{\text{GPT},n} = G(O_{n,1}, O_{n,2}, \ldots, O_{n,C})$$

This approach allows us to leverage GPT-4o's natural language understanding capabilities to assess the semantic consistency of the generated descriptions, providing a more nuanced evaluation than purely statistical methods.

Determining Preferred Image Distribution. Finally, we determine the preferred image distribution by comparing the consistency scores across all image distributions. The distribution with the highest average consistency score, considering all measurement methods (ROUGE/BLEU, BERT, and GPT-based), is considered the preferred distribution. This approach allows us to identify which image distribution leads to the most consistent and reliable outputs from the VLM.

## 3.2 Human Expert Rating and Dataset Creation

While consistency is a key indicator of model confidence, it is not sufficient on its own as the responses could be consistently incorrect. Therefore, we involve a mechanical expert to rate the explanations provided by the VLM for each part in different image distributions. The ratings focus on both accuracy and usefulness of the explanations. The overall expert rating results across all samples and explanations (i.e., 25 samples times 3 explanations for each) are summarized in Table 3. The criteria for expert ratings include *Relevance*, *Accuracy*, *Detail*, *Fluency*, and *Overall Quality*. We convert the options for these criteria to numerical values between 1 and 5 to calculate the values in Table 3. We also ask the human experts to add comments when necessary to provide additional insights.

The *relevance* and *accuracy* were evaluated by first analyzing the congruency between the name and the depicted image. A lower rating was assigned if the preliminary assessment revealed a lack of alignment. Subsequently, the rating was adjusted if the name and the content of the text did not align. A higher level of congruity indicated higher accuracy. From there, the contents were assessed for their ability to accurately describe the component design features, characteristics, industry, intended use, etc. The *detail* evaluation was assessed based on whether the provided data sufficed to conceptualize the design. *Fluency* was gauged by the grammatical correctness and the coherence of the descriptions. The *overall quality* was determined by the total of the scores from the indicated categories. While evaluating the different categories, an emerging trend was noticed. If the visual language model correctly identified the object's name, the subsequent details tended to align correctly. However, when the model misidentified the geometry, the details tended to correspond to the wrong item identification. For parts that were highly specialized for assembly, a more general example of industry standards was often indicated, rather than a specific standard as a starting point for further analysis by the end user.

From top-rated explanations, we developed a specialized dataset comprising CAD images paired with questions and answers extracted from the explanations. This dataset is designed to evaluate VLMs on visual question answering (VQA) tasks specific to CAD objects. By grounding our dataset in expert-validated explanations, we provide a reliable benchmark for assessing VLM performance in the CAD domain, bridging the gap between consistency and domain-specific accuracy.

## 4 CAD-VQA Dataset

We present CAD-VQA (Computer-Aided Design Visual Question Answering), a novel dataset designed to evaluate Vision-Language Models' understanding of 3D mechanical parts in CAD contexts.

### 4.1 Dataset Creation Process

Building upon the high-quality explanations generated through our iterative process of VLM output and human expert evaluation, we developed a novel dataset for evaluating Vision-Language Models on CAD tasks. The dataset creation process involved the following steps:

*Selection of top-rated explanations:* We chose explanations for 17 parts that received excellent ratings from human experts.

*Question generation:* Using Claude 3.5 Sonnet, we generated an initial set of questions based on these top-rated explanations. The questions cover various aspects including part names, geometrical features, assembly features, and functionality.

*Visual focus:* We designed questions to require analysis of the provided images, ensuring that answers couldn't be derived solely from common knowledge of 3D design.

*Comprehensive coverage:* A total of 85 multiple-choice questions were created, providing a diverse range of queries about the 17 selected parts.

*Quality assurance:* We conducted rigorous post-processing to ensure consistency in question style, eliminate errors, and maintain a uniform difficulty level across the dataset.

This dataset addresses a gap in the field of VLM evaluation for CAD applications. Currently, there is a scarcity of publicly available datasets specifically designed to assess VLMs' understanding of 3D mechanical parts and their features. Our dataset, while compact, represents one of the first efforts to create a benchmark for evaluating VLMs in the context of CAD and mechanical engineering.

The uniqueness of this dataset lies in its focus on:

- Specialized vocabulary and concepts from mechanical engineering and CAD

- Visual interpretation of 3D parts from multiple perspectives

- Understanding of both individual part features and their roles in larger assemblies

- Application of domain-specific knowledge to answer questions based on visual input

By providing this dataset, we aim to stimulate further research in improving VLMs' capabilities in specialized technical domains, particularly in the field of mechanical design and engineering.

To illustrate the nature of our CAD-VQA dataset, we provide a few representative examples in Table 1. These examples demonstrate the diversity of questions and the necessity of properly analyzing the provided images to correctly answer them.

| Image | Question and Options |
|---|---|
|  | **Q:** *Based on the visual representation, what role does the washer likely play in the assembly?* 

 A) It acts as a pivot point, B) It provides electrical insulation, C) It distributes the load of a fastener, D) It serves as a decorative element, E) It functions as a seal, F) It acts as a heat sink, G) It provides cushioning, H) It serves as a wear surface, K) It functions as a locking device, J) Both C and H are correct |
|  | **Q:** *Looking at the 2D images, which of the following names best describes this part?* 

 A) Gear Assembly, B) Piston Rod, C) Bracket Mount, D) Camshaft, E) Flywheel, F) Crankshaft, G) Bracket with Mounting Holes, H) Valve Cover, K) Both C and G are correct, J) Timing Belt |

Table 1: Sample data points from the CAD-VQA dataset

# 5 Results

For our preliminary experiments, we use a relatively small dataset due to the difficulty in scaling the rating process of detailed explanations by mechanical experts. Our dataset consists of 25 3D mechanical parts from the ABC collection [15], each part appearing within a larger assembly context. We evaluate four different image distributions for rendering these parts. *Distribution A*: Each part is rendered as an individual solid. *Distribution B*: Each part is rendered in the assembly along with other parts, where the other parts are transparent. *Distribution C*: Similar to Distribution B but slightly zoomed. *Distribution D*: A mix of Distributions A, B, and C (two samples from each).

These distributions were chosen to cover a range of contexts, although many other rendering methods are possible. For each part, we generate three different paraphrased prompts aimed at explaining the part's function and significance within the assembly. A sample of how the data looks is shown in Figure 1.

**Consistency Measurement.** We measure the consistency of the outputs using the methods described previously: ROUGE and BLEU scores, BERT embedding cosine similarity, and GPT-based consistency judgment. The results for each image distribution are summarized in Table 2. The results indicate that Distribution D, which includes a mix of the different rendering methods, consistently achieves the highest scores in both consistency metrics and expert ratings. This suggests that providing multiple perspectives of the parts helps the VLM generate more accurate and reliable explanations. Additionally, the use of in-context learning with expert feedback shows a noticeable improvement in the quality of the explanations, demonstrating the effectiveness of iterative refinement in enhancing model performance.

| Metric | Distribution A | Distribution B | Distribution C | Distribution D |
|---|---|---|---|---|
| ROUGE-1 | $0.4831_{\pm 0.0483}$ | $0.4479_{\pm 0.0606}$ | $0.4569_{\pm 0.0747}$ | $\mathbf{0.5159}_{\pm \mathbf{0.0609}}$ |
| ROUGE-2 | $0.1398_{\pm 0.0277}$ | $0.1298_{\pm 0.0369}$ | $0.1326_{\pm 0.0313}$ | $\mathbf{0.2055}_{\pm \mathbf{0.034}}$ |
| ROUGE-L | $0.2324_{\pm 0.0238}$ | $0.2267_{\pm 0.0307}$ | $0.2287_{\pm 0.0283}$ | $\mathbf{0.2916}_{\pm \mathbf{0.027}}$ |
| BLEU | $0.0874_{\pm 0.0176}$ | $0.0837_{\pm 0.0213}$ | $0.0865_{\pm 0.0173}$ | $\mathbf{0.1613}_{\pm \mathbf{0.0216}}$ |
| Cosine Similarity | $\mathbf{0.8988}_{\pm \mathbf{0.0289}}$ | $0.8902_{\pm 0.0401}$ | $0.8756_{\pm 0.055}$ | $0.8887_{\pm 0.041}$ |
| GPT Score | $0.6212_{\pm 0.2403}$ | $0.4365_{\pm 0.2716}$ | $0.4269_{\pm 0.2207}$ | $\mathbf{0.6308}_{\pm \mathbf{0.2504}}$ |
| **Average** | $0.4104_{\pm 0.0644}$ | $0.3691_{\pm 0.0769}$ | $0.3679_{\pm 0.0712}$ | $\mathbf{0.4490}_{\pm \mathbf{0.0723}}$ |

Table 2: Consistency scores across different image distributions. For all distributions, we randomly select 5 images rendered from various angles. For Distribution D ("All"), these 5 images are a mix drawn from the other three distributions.

## 5.1 In-Context Learning with Human Feedback

To further refine the model's performance, we use the expert ratings as feedback for in-context learning. The VLM is shown the expert ratings to learn and correct the explanations that received lower scores. After incorporating this feedback, we re-evaluate the model with human experts to assess improvement. The updated ratings are shown in Table 3.

Based on our consistency scores, Distribution D (a mix of single object renders, assembly renders with transparent parts, and zoomed assembly renders) performed best. We apply an in-context learning process to our dataset, using a prompt that provides the model with images, descriptions, and expert ratings for each part. The full in-context learning prompt can be found in Appendix A.3.

For our current dataset of parts, we provide GPT-4o with a comprehensive prompt containing all parts' information simultaneously:Iimages from Distribution D for each part, descriptions per part, and their corresponding human expert ratings. The model then generates new descriptions for parts based on this extensive in-context learning.

However, for larger datasets where providing all information at once may exceed the model's context length, we suggest two alternative approaches: a Sliding Window Approach and a Sequential Processing Approach. Details of these approaches and a visual comparison can be found in Appendix A.4.

| Metric | Before ICL-HF ↑ | After ICL-HF ↑ |
|---|---|---|
| Relevance | $3.88 \pm 1.34$ | **$3.96 \pm 1.27$** |
| Accuracy | $3.98 \pm 0.80$ | **$4.10 \pm 0.75$** |
| Detail | $4.14 \pm 0.69$ | **$4.16 \pm 0.68$** |
| Fluency | $4.06 \pm 0.75$ | **$4.10 \pm 0.73$** |
| Overall Quality | $4.07 \pm 0.79$ | **$4.14 \pm 0.73$** |

Table 3: Expert ratings across all samples and explanations before and after in-context learning. Score range is 1 to 5. ICL-HF refers to in-context learning with human feedback.

## 5.2 Performance of State-of-the-Art VLMs on our CAD-VQA dataset

We evaluated several state-of-the-art Vision-Language Models on our CAD-VQA dataset to establish baseline performance levels. The models tested include Claude-3.5-Sonnet [1], OpenAI's GPT-4o[21] and O1-preview, and Gemini-1.5-Pro [12].

Table 4 presents the accuracy of each model on our dataset:

| Model | Accuracy (%) |
|---|---|
| Claude-3.5-Sonnet | **61.17** |
| Gemini-1.5-Pro | 54.12 |
| GPT-4o | 54.11 |
| O1-preview | 42.35 |

Table 4: Performance of state-of-the-art VLMs on the CAD-VQA dataset

These results demonstrate that even the most advanced VLMs face significant challenges in accurately interpreting and reasoning about CAD objects. Claude-3.5-Sonnet shows the highest accuracy at 61.17%, while Gemini-1.5-Pro achieves 54.12% accuracy. These scores, while above random guessing (10% for 10-option multiple choice questions), indicate substantial room for improvement in VLMs' understanding of specialized technical domains like mechanical engineering and CAD.

The performance gap between these models and human experts underscores the need for continued research and development in enhancing VLMs' capabilities in domain-specific visual reasoning tasks.

## 6 Conclusion

Our study addressed the challenge of optimizing image distributions for black-box Vision-Language Models (VLMs). Experimenting with 3D mechanical parts and GPT-4o, we evaluated four image distributions using a novel methodology based on output consistency across paraphrased prompts. The mixed distribution, combining various rendering perspectives, consistently outperformed others, indicating that multiple viewpoints enhance VLM performance in generating accurate explanations. Expert ratings validated these findings and demonstrated the effectiveness of in-context learning with human feedback in improving explanation quality. Building on these insights, we developed CAD-VQA, a new dataset for evaluating VLMs on CAD-related visual question answering tasks. This dataset addresses a gap in the field and provides a benchmark for assessing VLM performance in specialized technical domains.

Our approach of automated consistency checks, followed by expert evaluation, offers a scalable method for assessing VLM outputs. The evaluation of state-of-the-art VLMs on CAD-VQA establishes baseline performance levels, highlighting both the potential and current limitations of VLMs in interpreting specialized visual data. While our experiments focused on CAD applications, this methodology and the principles behind CAD-VQA are broadly applicable to other domains requiring specialized visual interpretation. Future work should explore scaling this approach to diverse fields, applying the dataset creation process to other specialized domains, and investigating the relationship between output consistency and model confidence through comparison with explicit confidence estimation techniques and human evaluations.

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

# A Supplementary Material

## A.1 Paraphrasing Prompt

The following prompt was used to generate paraphrases for our experiments:

---

**Paraphrasing Prompt**

Please generate 3 paraphrases of the following prompt. Each paraphrase should maintain the same core meaning but vary in phrasing and complexity. Ensure a mix of minor variations (e.g., word order changes, synonym substitution) and more significant restructuring. The paraphrases should be diverse enough to test a language model's robustness to input variations, but not so different that they alter the fundamental query.

**Original prompt:**
"Please analyze the object shown in the image. Note that in some images, the 3D part might appear red when shown in an assembly format, while in others, it might look grey when presented as an individual part. Provide a detailed explanation of the object's name or type, its geometric features and shape, and its likely function or purpose within a larger system or assembly. Be as specific and comprehensive as possible in your description."

Generate your 3 paraphrases below:
1. [Paraphrase 1] 2. [Paraphrase 2] 3. [Paraphrase 3]

---

## A.2 Consistency Judgment Prompt

The following prompt was used for GPT-based consistency judgment:

---

**Consistency Judgment Prompt**

You are tasked with evaluating the consistency of multiple descriptions of the same 3D mechanical part. These descriptions were generated by an AI model in response to slightly different prompts about the same image. Your job is to assess how consistent these descriptions are with each other in terms of content, details, and overall interpretation of the part.
Please consider the following aspects:

1. Name/Type Consistency: Do all descriptions refer to the part using the same or very similar names/types?

2. Geometric Features Consistency: Are the descriptions of the part's shape, size, and key geometric features consistent across all versions?

3. Functionality Consistency: Do all descriptions attribute the same or very similar functions or purposes to the part?

4. Detail Level Consistency: Is the level of detail provided about the part similar across all descriptions?

5. Context Consistency: If the part's position or role within a larger assembly is mentioned, is this consistent across descriptions?

After analyzing the descriptions, please provide:

1. A consistency score from 0 to 1, where 0 means completely inconsistent and 1 means perfectly consistent.

2. A brief explanation (2-3 sentences) justifying your score.

Descriptions to evaluate: 1. [Description 1] 2. [Description 2] 3. [Description 3]
Your consistency score and explanation: [Score]: [Explanation]:

---

## A.3 In-Context Learning with Human Feedback Prompt

The following prompt was used for in-context learning with human feedback:

392

- Accurately identifies and names the part

- Describes its geometric features and shape in detail, referencing specific views from the five images

- Explains its likely function or purpose within a larger system or assembly

- Maintains consistency with the high-rated aspects of previous descriptions

- Improves upon areas that received lower ratings

- Integrates information from all provided perspectives

Your new description should aim to maximize all five rating categories: Relevance, Accuracy, Detail, Fluency, and Overall Quality.
Please provide your improved description.

## A.4 Alternative Approaches for Large Datasets

For larger datasets where providing all information at once may exceed the model's context length, we suggest two alternative approaches:

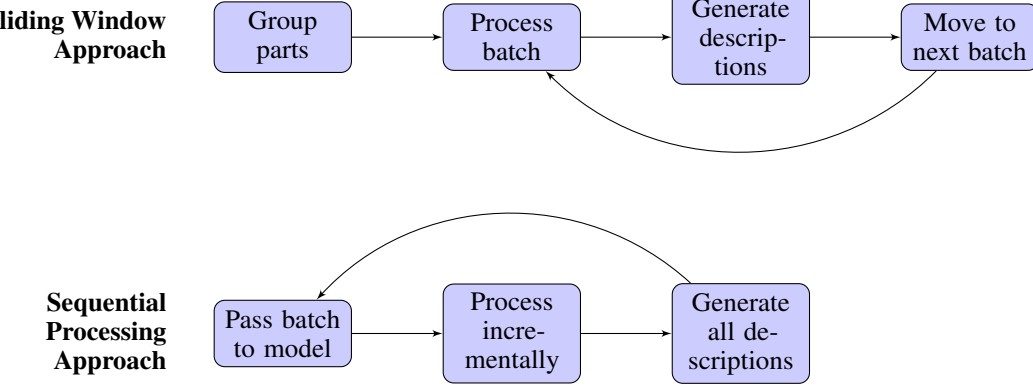

Figure 2: Visual comparison of algorithms for processing large datasets

These methods allow the model to learn from a substantial amount of context while remaining within practical limits. The Sliding Window Approach processes the data in overlapping batches, while the Sequential Processing Approach passes batches to the model incrementally before generating all descriptions at once.

