# OpenReview forum: "How to Determine the Preferred Image Distribution of a Black-Box Vision-Language Model?"
_NeurIPS.cc/2024/Workshop/SafeGenAi — SafeGenAi Poster_

### Official Review · Reviewer_T9Qx · 2024-10-08
**A Review on How to Determine the Preferred Image Distribution of a Black-Box Vision-Language Model?**

**Rating:** 6
**Confidence:** 4

**Review:**

This paper presents promising research but still has some room that needs more study.

The study shows a strong point in its new way of thinking. The authors have devised a novel method to check the confidence of VLMs by measuring output consistency across various image distributions using CAD images, without the need to access internal model parameters. Furthermore, they've introduced a new dataset for evaluating VLM performance on CAD tasks, addressing existing research gaps.

However, there are limitations to consider. Firstly, the dataset's small size of only 25 data points may introduce a degree of randomness to the results, potentially affecting their reliability. Secondly, the authors appear to have overlooked the possibility that the mixed distribution might provide additional details to the model, potentially leading to improved accuracy. This aspect deserves further investigation and discussion.

In conclusion, while this research offers valuable insights and tools for VLM evaluation, there's room for improvement in terms of dataset size and depth of analysis.

---

### Official Review · Reviewer_wn6T · 2024-10-09
**This paper proposes a novel method for determining preferred image distributions for Vision-Language Models (VLMs) using output consistency, introduces the CAD-VQA dataset for evaluating VLMs in specialized CAD tasks, and demonstrates the effectiveness of human feedback in improving model accuracy.**

**Rating:** 8
**Confidence:** 4

**Review:**

This paper proposes a novel method for determining preferred image distributions for Vision-Language Models (VLMs) using output consistency, introduces the CAD-VQA dataset for evaluating VLMs in specialized CAD tasks, and demonstrates the effectiveness of human feedback in improving model accuracy.

The paper is well-done, presents some new ideas, and could have a strong impact, especially in technical fields that rely on CAD

Pros:
- The paper introduces a novel method of using output consistency as a proxy for model confidence, which is a fresh and creative idea for black-box VLMs
- The introduction of the CAD-VQA dataset fills an important gap in evaluating Vision-Language Models in specialized technical fields like CAD, offering a valuable benchmark for future research
The use of in-context learning with human expert feedback adds a practical layer for refining model outputs and improving their relevance and accuracy

Cons:
- While the experiments are thorough, the discussion around the results could be more detailed, particularly in explaining the underlying reasons behind the performance of certain models. Is it because of lack of in-domain examples in training data? Or is something in the model architecture or training procedure contributing to that?
- High consistency in model outputs might not always correlate with accuracy, as consistent but incorrect outputs could mislead the evaluation of model performance. How can this be addressed?

---

### Official Review · Reviewer_gErd · 2024-10-09
**Novel work in the space of explainability of Generative VLM**

**Rating:** 8
**Confidence:** 3

**Review:**

The research paper presents novel ways to find the image distribution of unknown VLM, through the application of Computer-Aided Design (CAD).

Quality:
- Pros:
1. The work introduces evaluation methods for the generated responses using manual Human verification in the aspect of Relevance, Accuracy, Detail, Fluency, and Overall Quality.
2. Provides a broad evaluation metrics ROUGE/BLEU, BERT, and GPT- based scores.
3. The work also releases a CAD-VQA dataset which will benefit the community.
- Con:
The Image dataset distributions A, B, C and D has different levels semantic information embedded in it. I think it would be a important an evaluation after adding a normalization term to cancel out the additional contextual information encoded in the images, especially in Distribution B(where the part is rendered with the assembly) and Distribution D(mix of all other distribution). A model working with  more contextual information is bound to elicit a more relevant response.  Perhaps the Humans experts should also be evaluated on the same prompts and the variance of their responses can be used to normalize the variance of the reponses by the GPT model.

Clarity:
- Pros:
1. The work is explained clearly as it walks the reader through the limitations followed by a list of solutions to the forementioned limitations.

- Con:
1. The source and details of the human experts is not mentioned, providing this information on the experts will be useful in the context of comparisons.

Originality:
The work seems to be new and original in terms of CAD application.


Significance:
1. This work will be useful for the large community of CAD industry professional, students and researchers in academic instuitions.